# Early Neuroprotective Effects of Bovine Lactoferrin Associated with Hypothermia after Neonatal Brain Hypoxia-Ischemia in Rats

**DOI:** 10.3390/ijms242115583

**Published:** 2023-10-25

**Authors:** Eduardo Sanches, Yohan van de Looij, Dini Ho, Laura Modernell, Analina da Silva, Stéphane Sizonenko

**Affiliations:** 1Division of Child Development and Growth, Department of Pediatrics, School of Medicine, University of Geneva, 1205 Geneva, Switzerland; yohan.vandelooij@epfl.ch (Y.v.d.L.); dinihe0307@gmail.com (D.H.); laurammodernell@gmail.com (L.M.); stephane.sizonenko@unige.ch (S.S.); 2Center for Biomedical Imaging (CIBM), Animal Imaging and Technology Section, Ecole Polytechnique Fédérale de Lausanne (EPFL), 1015 Lausanne, Switzerland; analina.dasilva@epfl.ch

**Keywords:** hypoxia-ischemia, hypothermia, lactoferrin, brain metabolism, neuroprotection

## Abstract

Neonatal hypoxic-ischemic (HI) encephalopathy (HIE) in term newborns is a leading cause of mortality and chronic disability. Hypothermia (HT) is the only clinically available therapeutic intervention; however, its neuroprotective effects are limited. Lactoferrin (LF) is the major whey protein in milk presenting iron-binding, anti-inflammatory and anti-apoptotic properties and has been shown to protect very immature brains against HI damage. We hypothesized that combining early oral administration of LF with whole body hypothermia could enhance neuroprotection in a HIE rat model. Pregnant Wistar rats were fed an LF-supplemented diet (1 mg/kg) or a control diet from (P6). At P7, the male and female pups had the right common carotid artery occluded followed by hypoxia (8% O_2_ for 60′) (HI). Immediately after hypoxia, hypothermia (target temperature of 32.5–33.5 °C) was performed (5 h duration) using Criticool^®^. The animals were divided according to diet, injury and thermal condition. At P8 (24 h after HI), the brain neurochemical profile was assessed using magnetic resonance spectroscopy (^1^H-MRS) and a hyperintense T_2_W signal was used to measure the brain lesions. The mRNA levels of the genes related to glutamatergic excitotoxicity, energy metabolism and inflammation were assessed in the right hippocampus. The cell markers and apoptosis expression were assessed using immunofluorescence in the right hippocampus. HI decreased the energy metabolites and increased lactate. The neuronal–astrocytic coupling impairments observed in the HI groups were reversed mainly by HT. LF had an important effect on astrocyte function, decreasing the levels of the genes related to glutamatergic excitotoxicity and restoring the mRNA levels of the genes related to metabolic support. When combined, LF and HT presented a synergistic effect and prevented lactate accumulation, decreased inflammation and reduced brain damage, pointing out the benefits of combining these therapies. Overall, we showed that through distinct mechanisms lactoferrin can enhance neuroprotection induced by HT following neonatal brain hypoxia-ischemia.

## 1. Introduction

Neonatal hypoxic-ischemic encephalopathy (HIE) due to reduced oxygen and/or blood flow to the brain before or during birth is a high cause of mortality and morbidity [1,2]. Despite recent advances in neonatal intensive care, HIE still affects 1–3/1000 live births in high income countries and 2.3 to 30.6/1000 live births in middle and low incomes ones [3,4]. HI is considered an etiological factor, triggering HIE with subsequent diseases such as cerebral palsy (CP), epilepsy, autism (ASD), attention-deficit hyperactivity disorder (ADHD) and cognitive and neurosensorial deficits [5,6]. Since 2005, hypothermia (HT) has been the standard treatment for neonates with a gestational age (GA) above 35 weeks with moderate to severe HIE [7], and it is the only available strategy with beneficial effects for improving outcomes after moderate to severe brain lesions [8,9]. In this period, clinical trials have shown that HT reduces the risk of death in infants and significantly improves neurodevelopmental outcomes [10].

With the initial energetic collapse, the ionic pumps function is impaired, leading to Ca^++^ influx, membrane depolarization, glutamatergic excitotoxicity and oxidative stress [8]. Due to its high concentrations of free Fe^++^ and a low antioxidant system capacity, the immature brain is highly susceptible to the early consequences of energetic failure derived from the oxidative stress generated by multiple organelles, such as the mitochondria and endoplasmic reticulum [8,11]. Neuroinflammation, the systemic upregulation of pro-inflammatory cytokines, reactive astrogliosis [12,13] and microglial activation are also considered key features of HI injuries [14,15,16]. Experimental HI is valuable in order to better understand the mechanisms of damage and to test therapeutic agents. In regard to its complex and intricate mechanisms, no definitive cure for HI exists [17]. The Rice–Vannucci HI model in newborn rodents has been performed with success, providing insights into the mechanisms of injury, and it is a valuable model for testing therapeutic strategies. Its severity varies according to the parameters, such as the age at which it is performed, the sex of the animals and the intensity/duration of the hypoxia procedure [1,18,19,20] similar to HIE in newborns.

Treatments used for experimental HI target specific pathological injury processes, including excitotoxicity, oxidative stress, mitochondrial dysfunction and neuroinflammation [21]. Due to its success in protecting the developing brain following HI [19,22], HT became the standard therapy used for treating HIE patients. However, nearly 50% of infants do not benefit from the treatment, in part at least, due to the narrow window for starting the procedure [23,24,25,26]. HT has been shown to decrease metabolism and free radicals production, leading to a decrease in excitotoxicity and neuroinflammation. However, the mechanisms are not completely understood [9,22]. Despite the relative safety and effectiveness of decreasing the initial HIE damage, neuroprotection afforded by HT is incomplete and could be greatly benefited by adjuvant therapies [23,27,28,29]. In this sense, reducing the injury mechanisms and broadening the therapeutic window are major targets for such strategies.

Nutritional interventions can be an interesting approach for newborns at risk for HI events. Several nutritional agents have been shown to protect against HI injury by reducing the lesion infarct as well as reversing the behavioral impairments [30]. The mechanisms of action include antioxidant properties, such as those observed in resveratrol [31], melatonin [32] and pomegranate juice [33], to anti-inflammatory actions, such as following docosahexaenoic acid treatment [34,35]. In this context, we have shown that lactoferrin (LF), an iron-binding glycoprotein member of the transferrin family found in secretions (for review, see [36]), is a promising strategy for protecting the developing brain against injuries. LF is the main whey protein in breast milk [37,38,39] and its biological properties decrease HI and inflammation-induced brain damage [40,41] and reverse maturational delays following intrauterine growth restriction (IUGR) [42,43] through antioxidative, anti-inflammatory and neurotrophic properties. LF has been shown to protect the developing brain against HI damage in a dose response manner [44]. However, up to date LF has not been tested in the models of HIE with a term-equivalent brain age (P7, postnatal day), in which more severe tissue damage was observed in the cortical and subcortical grey matter areas, such as the hippocampus [45].

As such, the aim of this study is to test whether lactoferrin supplementation in the maternal diet given through lactation can increase the neuroprotective effects of hypothermia in a term-equivalent rat model of HIE early after injury. We evaluated the early in vivo brain neurochemical profiles using magnetic resonance spectroscopy (MRS) as well as the early neurodevelopmental reflexes impaired by HI. The mRNA levels of the genes related to excitotoxicity, energy substrate transport, astroglial reactivity and inflammation were assessed and the expression of the cellular markers and cell death in the right hippocampus 24 h after the HI procedure were also quantified. 

## 2. Results

### 2.1. Body Growth and Neurodevelopmental Reflexes

The animals submitted to HI had reduced body weight gain compared to the sham animals at P8 (24 h after injury) (F(5,89) = 6.71, *p* < 0.001) (Figure 1a). No effects of HT or LF were observed in this measure. Interestingly, the animals submitted to a LF diet presented a decreased body weight compared to the controls. The developmental milestones assessed at P8 did not show major deficits in the HI animals regarding the righting reflex (Figure 1b) or negative geotaxis (Figure 1c). However, a significant increase in the time to reach the home bedding was observed in the HINT animals (F5,78) = 2.37 *p* = 0.046) (Figure 1d), and a protective role of HT was present on this measure.

### 2.2. Brain Metabolic Profile Is Changed after HI with Partial HT and LF Effects

The neurometabolic profile measured by in vivo ^1H^MRS in the right hippocampus showed a decrease in glucose (F(5,42) = 2.05, *p* = 0.04) and Cr + PCr (F(5,44) = 17.54 *p* < 0.0001) (Figure 2a,b, respectively) that was partially reversed by HT alone. HI caused a significant drop in Cr + PCr; however, no effects of HT or LF were observed. Lactate, an index of a shift from aerobic to glycolytic metabolism, was prevented by HT alone and when LF and HT were associated (F(5,44) = 5.00 *p* = 0.001) (Figure 2c). There was an increase in Lac/NAA in the HI groups (F(5,42) = 2.84 *p* = 0.026) without evidence of enhancement due to the LF administration. This ratio is commonly used as a biomarker for neurological long-term brain injury [46], and HT alone mediated effective protection.

As depicted in Figure 3, the excitatory and inhibitory neurotransmission were affected by HI. Glutamate (F(5,44) = 4.12 *p* = 0.003), aspartate (F(5,44) = 3.78 *p* = 0.006) and GABA (F(5,42) = 6.24 *p* = 0.002) were decreased in the HI groups 24 h after HI with no effect from the treatments. Interestingly, the LFHINT animals presented increased levels of glycine (F(5,44) = 3.26 *p* = 0.01). Glycine acts as a neurotransmitter and NMDAr co-agonist, can mediate intracellular energetic processes and can be uptaken from the extracellular space by astrocytes and converted into lactate [47]. It is important to mention the decrease observed in aspartate (one of the main excitatory neurotransmitters in the CNS) [48] in the LFHIHT group (Figure 3b), which induced a decrease in the inflammation mediated by the interleukin-1β (IL-1β) production derived from inflammatory macrophages [49].

Macromolecules (F(5,44) = 9.53 *p* < 0.0001) as well as antioxidant metabolites were decreased due to HI with no evidence of recovery due to the treatments (Figure 4). Glutamine was increased in the HILF group (F(5,44) = 14.58 *p* < 0.0001), which suggested an increase in the astrocytic uptake of glutamate due to the LF administration that was reversed by HT, evidencing the competitive effects of the treatments. This could impact the neuron–astrocytic metabolic shuttle since synaptic homeostasis is highly dependent on the glutamatergic uptake.

The HI animals treated with lactoferrin presented no alterations in glutamate + glutamine, evidencing a preserved neuron–astrocyte function (Figure 5a). The glutamate/glutamine ratio reflecting neurotransmission and the glutamate–glutamine cycling between neurons and glial cells [50] were decreased. They were not reversed by any strategy. 

Additionally, the cell injury/viability markers such NAA (generally considered a marker for viable neurons) were decreased in all the HI groups (F(5,43) = 12.52 *p* < 0.0001). They were not reversed by any of the therapies when used alone or in combination (Figure 6a) during this early assessment.

### 2.3. HT and LF Have Distinct but Complimentary Pathways for Neuroprotection following HI

After the energetic failure caused by HI, glutamatergic excitotoxicity, oxidative stress and inflammation are common features. In order to assess this aspect, we looked at the transcription levels of the genes involved in these mechanisms that could be modified by the therapies alone or in combination. 

Excitotoxicity, evaluated by the transcription levels of GRIN2a (NMDA2r receptor) (F(5,39) = 9.54 *p* < 0.0001) was decreased in all the HI groups, indicating significant cell death with no evidence of protective effects from any of the therapies (Figure 7a). Interestingly, there was an increase in the mRNA levels of slc1a3 (GLAST glutamate transporter) in the LFHINT group (F(5,40) = 4.56 *p* = 0.002) (Figure 7b), which could indicate an increased vulnerability to HI in the animals submitted to a LF diet that was prevented by HT.

Following HI, transcription factors such as HIF-1α respond to hypoxic conditions leading to an increased macrophage activation and expression of pro-inflammatory cytokines in the ipsilateral hemisphere to carotid ligation. The hippocampus was highly vulnerable to HI conditions at this developmental stage [11,17]. LF was more effective in counteracting the increase in HIF-1α (Figure 8a). The combination of LF and HT was effective in reversing the increase in TNF-α (Figure 8b). The animals fed with LF had increased IL-1β levels, which was not observed in the HINT animals. 

One hallmark of HI injury is reactive astrogliosis [51]. We observed an increase in the astrocyte mRNA levels of GFAP in all the HI groups with no decrease due to any treatment (F(5,37) = 12.92 *p* < 0.0001) (Figure 9a). However, S100B, considered an important biomarker of HI brain injury and activated when astrocytes are reactive, was increased in the HI groups and returned to the sham level only when HT and LF were combined (F(5,38) = 6.73 *p* = 0.0001) (Figure 9b).

The failure in energy metabolism observed in vivo through ^1H^MRS highlights the impact of HI in the glucose and lactate transport across the cells. In this sense, the GLUT-1 mRNA and lactate transporter HCAR1 mRNA were increased following HI, evidencing the hippocampus’ need for energetic substrates after injury (Figure 10a,b). Interestingly, HT protection seemed to be related to the glucose metabolism failure prevention and LF increased the lactate transport into the astrocytes, possibly to support the glutamate uptake. 

Since we observed distinct metabolic pathways for each strategy, we looked into the neuronal and glial energy transporters. Once again, MCT-2, which was present in the neurons, was decreased only in the HINT group (F(5,39) = 2.63, *p* = 0.03) (Figure 11a), suggesting HT and LF administered alone or combined prevented the neuronal energetic failure. Further, we observed distinct but complimentary effects of LF and HT over the glial transporters MCT-1 and MCT-4. While MCT-1 (which facilitated the transport of lactate from the neurons into the astrocytes) was increased due to HIHT (F(5,34) = 3.17 *p* = 0.0186), MCT-4 (mainly responsible for the release of lactate from astrocytes) (F(5,39) = 2.69 *p* = 0.03) was increased in the LFHINT animals (Figure 11a,c). 

### 2.4. Combination of LF and HT Decreases the T2W Hyperintense Signal 24 h after HI

The MRI analysis revealed an increase in the brain volume of the HI animals fed with a control diet (F(4,56) = 2.62 *p* = 0.04), evidencing the tissue swelling characteristic of HI. This was not observed in the lactoferrin-fed animals or in the HI animals submitted to hypothermia (Figure 12a). Damage was present in all the HI animals; however, when lactoferrin was associated with hypothermia, the damage volume (based on the hyperintense signal) was significantly reduced, evidencing the synergy between the therapies (F(5,68) = 11.58 *p* < 0.0001) (Figure 12b).

### 2.5. HT Prevented CA1 Apoptosis with No Additive Effects of LF following HI 

We assessed the CA1 hippocampus damage by evaluating the immunofluorescence of the expression of GFAP, Iba-1 and HIF-1α (Figure 13). A trend of an increased expression of GFAP (F(5,23) = 1.93, *p* = 0.12) and Iba-1 (F(5,30) = 1.93, *p* = 0.11) was observed in the HI animals, evidencing the pro-inflammatory profile of the model. HIF-1α (F(5,28) = 2.53, *p* = 0.04) was increased in the HINT animals (F(5,28) = 6.15, *p* = 0.0006)(Figure 13c), which was in agreement with the increase in the MCT-1 mRNA levels observed in the same group.

Additionally, double staining for NeuN and cleaved caspase-3 was performed in the region (Figure 14). However, an increase in apoptosis in CA1 suggested by the cleaved caspase-3 expression (F(5,26) = 4.93, *p* = 0.002) in the HINT group was observed (Figure 14b), suggesting a more pronounced effect of LF for reducing apoptotic cell death after HI.

## 3. Discussion

Early metabolic dysfunction caused by neonatal HI is one of the main targets in the search for neuroprotective strategies [8,9]. Hypothermia has been shown to counteract the initial energetic failure when delivered in the initial phase of injury; however, from the point of view of logistics, it is not always possible. Thus, adjuvant therapies aiming to broaden hypothermia therapeutic effects or increase the protection in early stages after injury are urgently needed [29]. We showed that bovine lactoferrin administered through lactation for a short period (48 h) could induce neuroprotection and enhance the therapeutic effects of hypothermia in a model of neonatal HIE. HI caused marked brain metabolism decreases in the right hippocampus and mild to severe brain injury in its sub-acute phase (24 h after the procedure), as was observed with in vivo 1H-MRS. Overall, while hypothermia seemed to have a role related to glucose metabolism neuronal functioning maintenance, lactoferrin had effects related to astroglial function and, when combined, the therapies increased their therapeutic potential, preserving early tissue damage to a greater extent.

The early evaluation of the developmental milestones after neonatal injuries can give valuable information of the efficacy of the therapeutic strategies [52,53,54]. We observed a general decrease in body weight in the animals fed with a LF diet, which could be interpreted as a detrimental effect of the diet. Interestingly, when performed at P3, HI caused a decrease in body weight (as observed here) that was not prevented by LF in the long term [44]. However, previous studies reported that, in the models of energy restriction, LF supplementation enhanced weight loss [55] and other studies showed that the LF concentration was inversely associated to the body mass index [56]. Despite a trend of a worsening performance in the HINT group for righting the reflex and negative geotaxis, no statistical difference was observed. This was in agreement with the previous data that did not observe early impairments after HI [57,58]. Another important milestone that was assessed was the olfactory recognition test. We observed an improvement in the HI animals receiving HT, with no additive effects of lactoferrin in the task pointing out the efficacy of HT for reversing this early behavioral impairment. The better performance in the task, despite primarily testing the olfactory function, was also dependent on the improved motor function. Our data suggested that the preservation of the motor ability using HT may have had an impact on other sensorial domains and helped HI animals. In agreement, despite no evidence of improvement due to HT in the early reflex testing after HI, Diaz and colleagues observed long-term motor improvements in the open field [54]. 

HI leads to primary brain energetic failure, lasting from minutes to a few hours [59,60]. Imaging techniques such as MRI and MRS are valuable tools for detecting such lesions and providing prognostics according to the severity of the tissue damage [61,62]. ^1^H-MRS allowed for an in vivo assessment of the early alterations in the brain metabolism due to neonatal HI in preclinical and clinical setups with high accuracy [61,62,63,64,65]. The brain metabolite concentrations had marked alterations 24 h after injury and the hyperintense signal on the T_2_W images evidenced the mild to severe characteristics of the brain damage observed in the study [46]. HI caused a decrease in the energetic metabolites (namely glucose, Cr and PCr) and the excitatory and inhibitory neurotransmitters. A glucose drop due to HI was prevented by HT, but in combination with LF, the effect was abolished, evidencing the competitive effects between the therapies which must be considered. The alterations induced by HI in NAA (an early marker of neuronal damage) were not attenuated either by HT or LF or in combination. However, lactate accumulation was prevented when both strategies were applied together, which highlighted the potential of using lactoferrin as an adjuvant therapy to HT. The ratio of Lac/NAA, which is frequently used as a biomarker of long-term neurological sequelae of HI [46], was increased in the HINT groups as expected and reduced in the HT groups (HIHT and LFHIHT) with no effects from LF alone. This provided evidence of the more pronounced effects of HT during early energetic failure and had a prominent role in neuronal survival. Regarding neurotransmission, as expected, there was an overall decrease in the inhibitory and excitatory hippocampal functions due to HI. Contrary to our initial hypothesis, the glutamate decrease seemed to be less intense in the lactoferrin-treated animals. When used together, the therapies did not reverse such impairment, further highlighting the possible competitive effects of combining the therapies. 

^1^H-MRS gave us the first clue that following HI, lactoferrin had an important effect on astrocytic function. Glutamine was increased in the LFHINT group, which could have been an attempt at removing glutamate from the synaptic cleft by the astrocytes and maintaining synaptic homeostasis [66]. Recent studies depicted the role of LF on astrocyte functioning and its implication on neurodevelopment and neurodegenerative diseases [67,68,69]. Xu and colleagues 2022, observed that the reduction in the endogenous LF levels in the astrocytes resulted in a selective reduction in cholesterol synthesis in the brain, impairing neuronal outgrowth and the establishment of synaptic structures during development [70]. Glutamine was also involved in mitochondrial oxidative phosphorylation and, consequently, in energy production for the cells [71]. High hippocampal Gln levels (particularly in situations of decreased Glu/Gln ratio) could be regarded as a multifunctional metabolic reservoir that, depending on the cellular requirements, could be readily utilized for fueling mitochondrial function. Indeed, our results suggested that lactoferrin acted directly via astrocytes, improving their role in maintaining metabolic storage in case of mitochondrial dysfunction caused by HI. This was supported by the observed increase in the mRNA levels of the astrocytic transporters of the substrates slc1a3 (or GLAST), HCAR1 and MCT-4 in the LFHINT animals.

After observing the metabolic disruption in the hippocampus due to HI and the protective effects of the HT and LF treatments using MRS, we asked if the early markers of oxidative injury could be modified by the treatments. An HIF-1α increase indicated a clear disbalance in excitatory transmission, leading to oxidative stress and cell death, which was attenuated by the combined therapies and could reduce the initial damage and further apoptosis [13]. Similar to this, HT induced an increase in the expression of HIF-1α, which could be due to the increase in the transfer of lactate from astrocytes to neurons via MCT1 transporters [72]. Interestingly, LFHINT increased the TNF-α and IL-1β mRNA levels compared to SH, which could indicate a different vulnerability to HI in the animals fed with LF, but could also reflect an accelerated inflammatory response in order to solve tissue distress and damage [73]. Interestingly, recent data showed that IL-1β could promote positive outcomes and reduce neuronal cell death via the stimulation of neurotrophins [74,75,76]. Together, the decreased expression of cleaved caspase-3 in all the treated groups using an immunohistological assessment pointed out the possible complimentary effects of combining the therapies, despite evidence showing that even alone, HT had important anti-apoptotic neuroprotective effects [77]. 

Using T_2_W images, we observed an increased volume in the right hemisphere in the HINT animals, which was indicative of cell swelling. Another important measure, the brain damage area was assessed using the hyperintense signal to measure the lesion volume. We observed that the combination of lactoferrin and hypothermia had a synergistic effect, as it was solely able to reduce acute brain damage. Importantly, this measure has been shown to preserve the correlation with long-term brain damage; however, a long-term assessment was not in the scope of the study and must be further addressed by the group [40]. In agreement, the recent data has shown the HT alone did not significantly reduce brain damage; however, when administered in combination with umbilical cord blood cells it reduced the astrocyte reactivity and lesion volume [59]. This reinforces that HT alone was not fully protective and that its efficacy on reducing lesion size may increase when combined with other neuroprotective agents, such as lactoferrin. The histological assessment of hippocampal brain damage showed that there was a clear trend for upregulated astroglial reactivity after HI that seemed be suppressed to a greater extent by HT, which was in agreement with the literature [59]. Neuronal apoptosis was prevented by HT and LF when administered alone and presented no synergetic effects when both treatments were administered. 

The study had some limitations that need to be considered. First, we analyzed a single early timepoint after injury (24 h) and did not continue for long-term behavioral and histological assessments. However, this first data set was important due to its capacity for showing that, through distinct pathways, LF and HT can interact and increase neuroprotection. Certainly, long-term neuropathological and functional evaluation of the treatments are needed to confirm this potential. Second, while we did not assess the protein levels for all the targets assed by the RT-qPCR, our goal was not to quantitively estimate the changes using proteomics, but rather, to observe how the HI model and interventions would lead to cellular responses. For this purpose, the literature clearly supported that transcriptomics could be a useful tool for evaluating these processes, since there was a good level of correlation between the changes in the mRNA and proteins. In CNS research, the adoption of transcriptomics as a tool for evaluating astrocyte and microglial phenotypes has also been increasing to a point of becoming the norm [60,78,79].

In conclusion, our data suggested that neuroprotection induced by HT could be boosted by the use of oral Lf in order to counteract the primary phase of injury, reduce glutamatergic excitotoxicity and oxidative stress and provide metabolic support, which in turn could have the potential for decreasing apoptosis and inflammation mitigating brain damage. It is important to highlight that despite all the evidence of synergistic neuroprotective effects between HT and LF in the context of HI, there were also non-synergetic effects that showed that a combination of hypothermia and lactoferrin therapies needs a precise evaluation for its effects, mechanisms of action and sequence of administration. The lactoferrin effects were related to a reduced astrocyte reactivity and inflammatory response, which was in agreement with the previous findings that showed LF as an immunomodulatory protein. The metabolic alterations, however, were mostly influenced by hypothermia protocol, which when combined seemed to be promising for increasing tissue protection. We further need to investigate the reasons why the combination of both therapies had some antagonistic effects and which mechanisms could be involved. Therefore, the search for adjunctive effective therapeutic interventions in addition to hypothermia continues. Here, we showed that lactoferrin has this potential.

## 4. Materials and Methods

### 4.1. Animals

The Geneva State Animal Ethics Committee and Swiss Federal Veterinary Service approved the study as VD3676. Pregnant Wistar rat dams were purchased from Charles River, France and, to reduce the inter-variability within the litters, all the groups were processed in parallel for each set of experiments. The experiments were carried out in standard animal housing conditions (22 ± 2 °C, 12 h light/dark cycle) with free access to food and water. On the first day of life (P0), the rat pups were sexed, and the litters were limited to 10 pups. Animals from the same litter were used for different assessments in order to avoid “litter effects”. The experimental timeline is presented in Figure 15. 

### 4.2. Brain Hypoxia-Ischemia 

HI was performed using the Rice–Vannucci HI model at P7, as previously described [40,46]. Briefly, 7-day-old male and female rat pups (1:1 ratio) underwent unilateral right common carotid artery occlusion under anesthesia using isoflurane (4% induction and 1.5% maintenance). The procedure lasted less than 5 min per animal. After a 90 min recovery period (in which the animals were returned to their mothers), the rats were submitted to a hypoxic atmosphere (8% O_2_ balanced in 92% N_2_) at 37 °C for 60 min. In general, at this developmental stage, the parameters used in this study caused a moderate degree of injury [80] and they were chosen since hypothermia may not be effective in cases of severe HI [81,82]. The sham-operated rats had neither a common carotid artery ligation nor hypoxia exposure. 

### 4.3. Therapeutic Strategies

In the lactoferrin groups, the dams were fed a diet supplemented with bovine lactoferrin (LF) (Apolactoferrin/LPS-free, Taradon laboratory, Tubize, Belgium) at a dose of 1 g/kg (Provimi Kliba SA, Penthalaz, Switzerland) from P6. The control dams were fed an isocaloric diet using casein to remain isocaloric with the LF diet, as previously described [44]. At P7, immediately after hypoxia exposure, the HT groups were submitted to whole-body hypothermia using a Criticool^®^ device. HT consisted of a period of cooling for 30 min, in which the animals were cooled to a minimum of 33.5 °C head temperature; a period of a maintained HT target temperature of 32.5–33.5 °C for 5 h; and a period of rewarming for 30 min, in which the animals reached 36–37 °C—normothermia. The temperature was measured on the head (“scalp temperature”) using an infrared thermometer (TFA SCANTEMP 330^®^) every 10 min during Phase I (cooling) and Phase III (rewarm) of HT (initial and final 30 min). During Phase II (HT properly), the animals’ head temperature was assessed every 30 min [52]. The average temperature during the hypothermia procedure is shown in Figure 16. 

### 4.4. H-Magnetic Resonance Spectroscopy (MRS) and T_2_WI Lesion Volume Assessment

The magnetic resonance experiments were performed on an actively shielded 9.4 T/31 cm magnet (Agilent/Varian/Magnex) equipped with 12-cm gradient coils (400 mT/m, 120 µs). In order to observe the brain neurochemical profile following HI, the animals were scanned at P8 (i.e., 24 h post HI) using a homemade quadrature transmit–receive surface radiofrequency coil with a 17-mm diameter, as previously described [40]. In short, the animals were placed under isoflurane anesthesia (1.5–2.0%) in supine position in a homemade holder. Their body temperatures (37 ± 0.5 °C) and heart rates were monitored and regulated during the MR acquisition [83]. A fast spin echo T_2_W image was performed to position the ^1^H-MRS voxel of interest. The ^1^H-MRS spectra acquisition was performed on the right hippocampus (voxel of interest of 1.5 × 1.5 × 2.5 mm^3^) using an ultrashort echo time (TE/TR52.7/4000 ms) SPECIAL spectroscopy method. The spectra were acquired in 16 blocks of 16 averages after automatic FASTMAP shimming [84]. The LC model [85] was used to analyze the acquired spectra and quantify the aspartate (Asp), ascorbate (Asc), creatine (Cr), phosphocreatine (PCr), g-aminobutyric acid (GABA), glucose (Glc), glutamate (Glu), glutamine (Gln), glycerophosphocholine (GPC), glycine (Gly), lactate (Lac), taurine (Tau), macromolecules (Mac), N-acetylaspartate (NAA), N-acetylaspartylglutamate (NAAG) and phosphocholine (PCho). 

### 4.5. Neurodevelopmental Reflex Testing 

In order to assess the functions in the initial stage following HI injury, the animals were tested for righting and negative geotaxis reflexes and olfactory memory test at PND8 (24 h after injury), as previously described [53]. In the righting reflex testing, the pups were placed in supine position and the time, in seconds, to turn them over to prone and place all four paws on the surface was recorded at a maximal time of 10 s. Negative geotaxis provided information about the vestibular and proprioceptive functions. It consisted of an automatic and stimulus-bound orientation movement. The animals were placed head down in the middle of an inclined 30 cm board (30° angle) and the latency to make a 180° turn was recorded in a period of 1 min. If an animal did not complete the test, 60 s was assigned. In order to assess early memory processing, olfactory discrimination was performed. The animals were placed in the center of a clean box (20 × 40 × 20 cm) with soiled bedding from the home cage placed at one side and the same quantity of clean bedding on the contralateral side of the box. The latency to reach the home bedding was recorded at a maximal time of 180 s. If the animal chose the clean bedding or did not reach the clean one, 180 was attributed.

### 4.6. Tissue Harvesting and Processing

For the RT-qPCR analysis, the animals were decapitated at P8 and the right hippocampi were quickly harvested and dissected on ice, frozen in liquid nitrogen and kept at −80 °C until analysis. For immunofluorescence, the animals were deeply anesthetized using sodium thiopental (150 mg/kg; i.p) and transcardially perfused with PBS followed by 4% paraformaldehyde (PFA; Sigma-Aldrich, St. Louis, MO, USA) in a 0.1 M phosphate buffer saline (PBS) pH 7.4. The brains were removed from the skull and post-fixed overnight. Then, they were dehydrated in 30% sucrose and frozen in isopentane at −20 °C for posterior slicing using a cryostat (Thermofischer—*NX50*, Waltham, MA, USA).

### 4.7. RT-qPCR

The hippocampal mRNA extraction was done using the RNase Mini Kit (Qiagen, 74104, Hilden, Germany), following manufacturer’s instructions. Briefly, 3 μg of mRNA from the right hippocampus was reverse transcripted onto cDNA using 400 units of Moloney murine leukemia virus reverse transcriptase (Invitrogen, 28025-013, Waltham, MA, USA), 20 units of recombinant RNAsin (Promega, N2515, Madison, WI, USA), 0.5 μg of random hexamers (ThermoFischer Scientific, #S0142), 2 mmol/L dNTP (Invitrogen, 10297018) and 40 mmol/L of dithiothreitol (Invitrogen, 18080093), as previously described [42]. A real-time quantitative PCR was performed using the PowerUp SYBR Green Master Mix (Applied Biosystems, A25742, Waltham, MA, USA) and the StepOne Plus^®^ Real-Time PCR System (Applied Biosystems). The gene expression was normalized using GAPDH as the housekeeping gene, relative to the SH group (considered as 1). The results were calculated using the 2^−ΔΔCT^ method and were expressed as the mean ± SE of the arbitrary units (AU). The sequence of bases used for the genes analyzed is described in the Appendix A.

### 4.8. Immunofluorescence

The coronal 25 μm brain slices apart 300 μm were collected from four to six animals per group. The P8 brains dorsal hippocampi (CA1) were assessed from -1.20 to -1.60 mm Bregma levels, according to the brain coordinates from the developing brain atlas [86]. The glial fibrillary acidic protein (GFAP) (astrocytic marker; 1:500, 6171—Sigma-Aldrich), NeuN (neuronal marker; 1:300, MAB377—Chemicon International, Inc., Temecula, CA, USA), Iba-1 (macrophage—microglial marker; 1:250, ab5076—abcam, Cambridge, UK), cleaved caspase-3 (apoptosis; 1:300, G6671—Cell Signaling, Cambridge, UK) and HIF-1α (hypoxia-inducible factor; 1:250 1 ab179483) were quantified. Briefly, the slides were washed in PBS and blocked for 60 min using a 10% goat or donkey serum in PBS with 0.3% Triton X-100 at room temperature. Then, the slides were incubated overnight with the primary antibody at 4 °C in PBS, 0.3% Triton X-100 and 3% of the same serum (donkey or goat). On the following day, the slides were washed in PBS, incubated in the dark with secondary fluorescent antibodies for 2 h at room temperature and mounted and cover-slipped with anti-fading acidic mounting media containing DAPI (Fluorsave^®^—Sigma). The secondary antibodies Alexa Fluor 488, 555 or 647 were used (Alexafluor^®^, ThermoFischer Scientific). The primary antibody was omitted on some sham animal slides for immunostaining negative controls. The images were acquired using a Nikon Eclipse T2000 microscope (Nikon^®^, Tokyo, Japan) coupled to a DS-Qi2 camera with a 10x objective. The average optical density intensity (ODI) of three regions of interest (ROI) of 400 μm^2^ in the right hippocampus CA1 were assessed in two consecutive slides. The ODIs were quantified using the NIH-ImageJ software (version 1.8.0) by an experimenter blinded to the groups. 

### 4.9. Statistics

The statistical analysis was performed using the SPSS™ software version 21 or Graphpad. The animals were obtained from at least three different litters and were chosen randomly for each technique. The normally distributed data was analyzed using a one-way ANOVA followed by Tukey’s *post hoc*. The Kruskal–Wallis test followed by the Mann–Whitney test were performed in case of a non-normal distribution. The results were presented as the mean ± SEM. A significance level was accepted when *p* < 0.05.

## Figures and Tables

**Figure 1 ijms-24-15583-f001:**
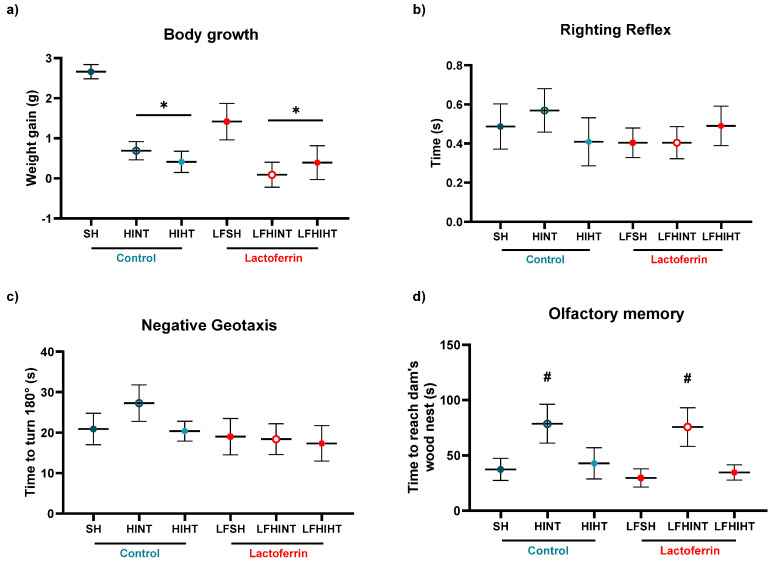
Early weight gain and neurodevelopmental milestones. HI reduced body growth in all the experimental groups. No effects of HT or LF were observed. No differences were observed in (**a**) the righting reflex or (**b**) the negative geotaxis due to HI or negative geotaxis (**c**). (**d**) Olfactory memory testing showed that HT reversed the impairment to reach the mother’s nest in the HI animals. The data are presented as the mean ± SE (the dots represent the average of 11–19 animals/groups). * vs. SH, ^#^ vs. LFSH.

**Figure 2 ijms-24-15583-f002:**
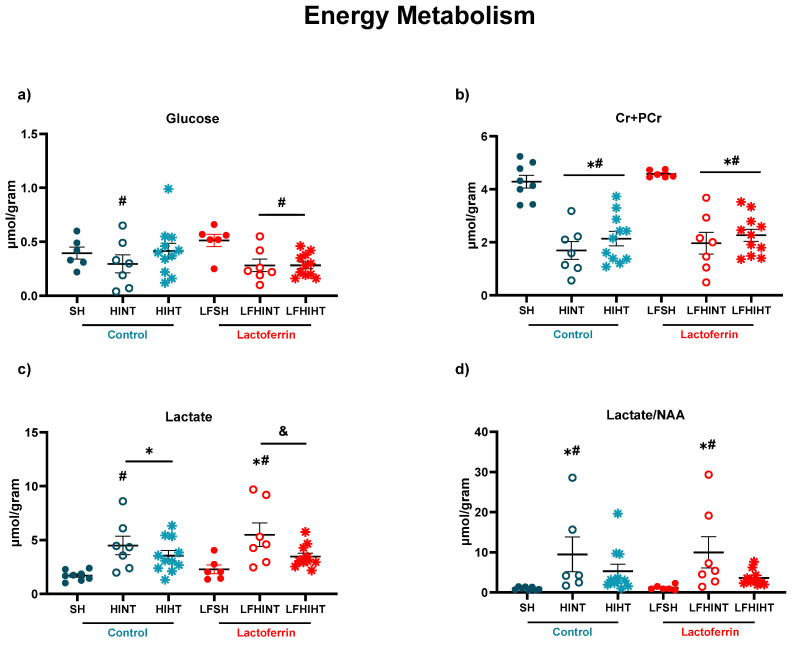
^1H^MRS shows HI caused a decrease in the metabolites related to energy metabolism of Glucose (**a**), Cr+PCr (**b**), Lactate (**c**) lactate/NAA ratio (**d**) in the right hippocampus 24 h after HI. The data are presented as the mean ± SE. A one-way ANOVA was followed by Tukey’s *post hoc* * vs. SH ^#^ vs. LFSH ^&^ LFHINT vs. LFHIHT.

**Figure 3 ijms-24-15583-f003:**
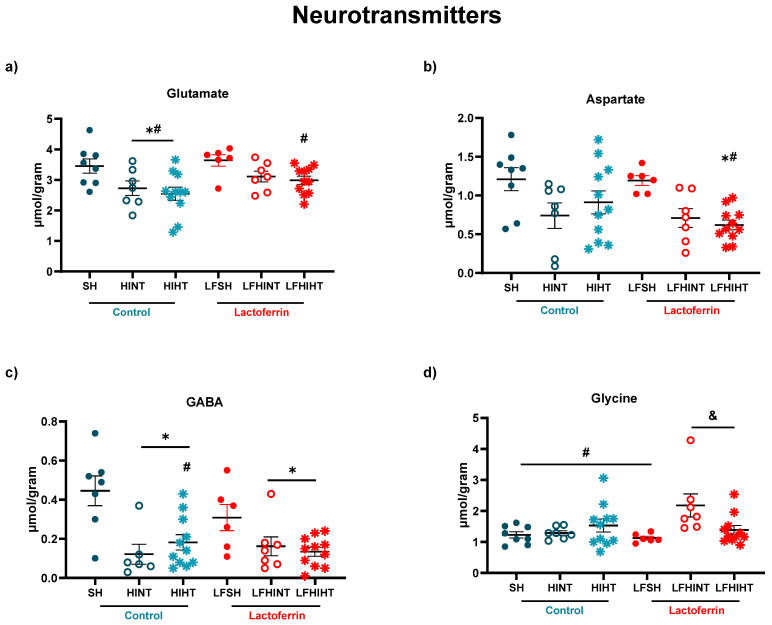
^1H^MRS assessment of the metabolites related to neurotransmitters glutamate (**a**), aspartate (**b**), GABA (**c**) and glycine (**d**) in the right hippocampus 24 h after HI. The data are presented as the mean ± SE. A one-way ANOVA was followed by Tukey’s *post hoc* * vs. SH ^#^ vs. LFSH ^&^ LFHINT vs. LFHIHT.

**Figure 4 ijms-24-15583-f004:**
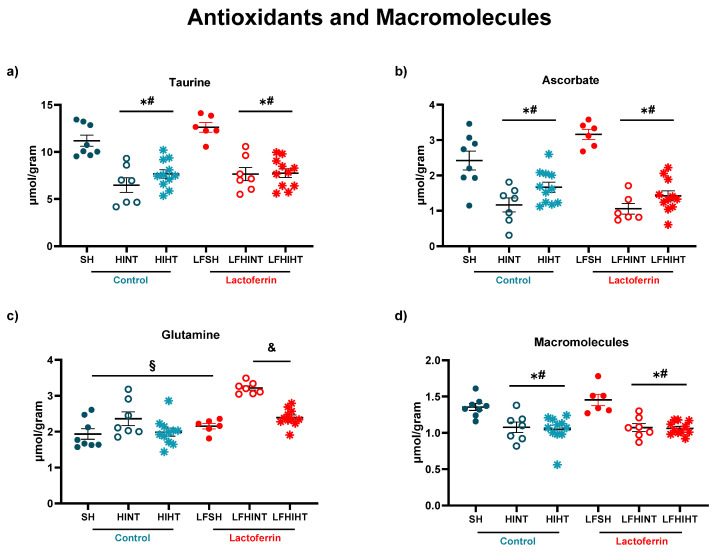
^1H^-MRS assessment of the antioxidants and macromolecules in the right hippocampus 24 h after HI. Hippocampal concentrations of Taurine (**a**), Ascorbate (**b**), Glutamine (**c**) and Macromolecules (**d**) 24h after HI. The data are presented as the mean ± SE. A one-way ANOVA was followed by Tukey’s *post hoc* * vs. SH, ^#^ vs. LFSH, ^§^ vs. LFHINT, ^&^ LFHINT vs. LFHIHT.

**Figure 5 ijms-24-15583-f005:**
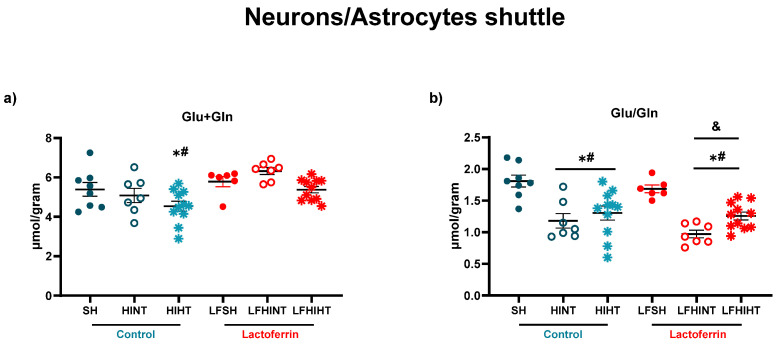
^1H^MRS assessment of the metabolites related to neuron/astrocyte coupling in the right hippocampus 24 h after HI (**a**) Glu+Gln (**b**) Glu/Gln. The data are presented as the mean ± SE. A one-way ANOVA was followed by Tukey’s *post hoc* * vs. SH, ^#^ vs. LFSH, ^&^ LFHINT vs. LFHIHT.

**Figure 6 ijms-24-15583-f006:**
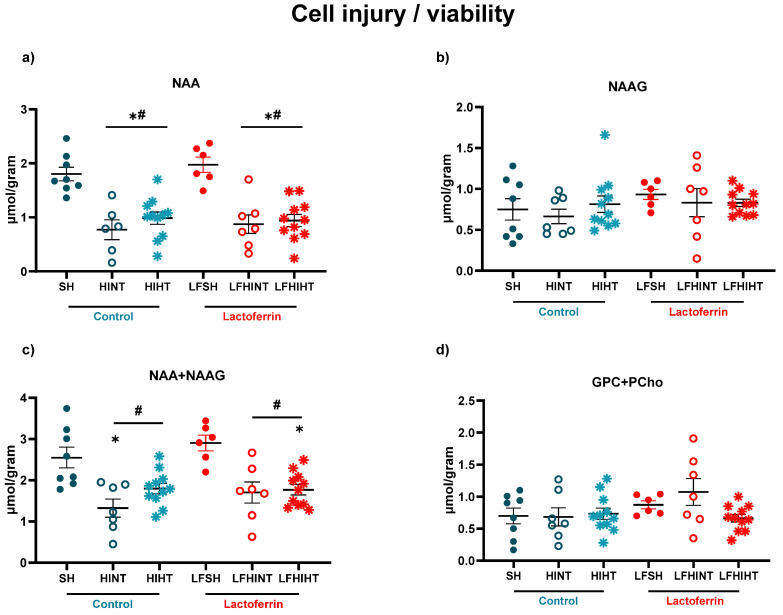
^1H^MRS assessment of the metabolites representative of cell injury and cell viability in the right hippocampus 24 h after HI. Hippocampal concentrations of N-acetyl-aspartate (**a**) N-acetyl-aspartyl-glutamate (NAAG) (**b**), NAA+NAAG (**c**) and GPC+PCho (**d**). The data are presented as the mean ± SE. A one-way ANOVA was followed by Tukey’s *post hoc* * vs. SH, ^#^ vs. LFSH.

**Figure 7 ijms-24-15583-f007:**
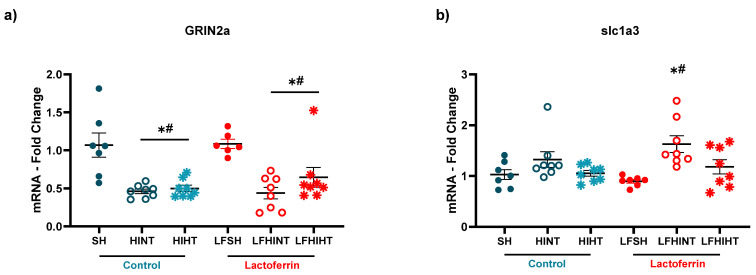
mRNA levels of the genes regulating excitotoxicity NMDAr2A (GRIN2a, in (**a**)) and glutamate uptake (slc1a3-GLAST, in (**b**)) in the right hippocampus 24 h after HI. The data are presented as the mean ± SE. A one-way ANOVA was followed by Tukey’s *post hoc* * vs. SH, ^#^ vs. LFSH.

**Figure 8 ijms-24-15583-f008:**
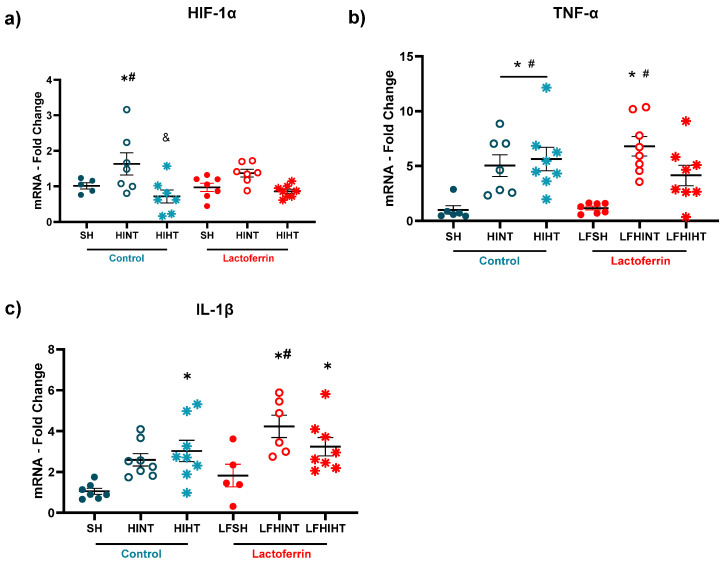
mRNA levels of the genes activated by hypoxia (HIF-1α) (**a**) and inflammatory markers TNF-α (**b**) and and IL-1β (**c**) in the right hippocampus 24 h following HI. The data are presented as the mean ± SE. A one-way ANOVA was followed by Tukey’s *post hoc* * vs. SH, ^#^ vs. LFSH.

**Figure 9 ijms-24-15583-f009:**
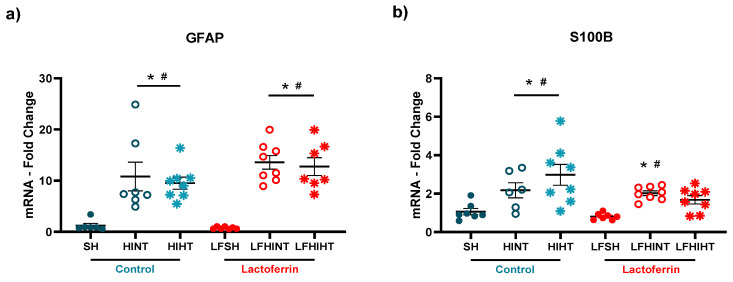
mRNA levels of the genes related to astrocyte reactivity (GFAP (**a**) and S100B (**b**), respectively) in the right hippocampus 24 h after HI. The data are presented as the mean ± SE. A one-way ANOVA was followed by Tukey’s *post hoc* * vs. SH. ^#^ vs. LFSH.

**Figure 10 ijms-24-15583-f010:**
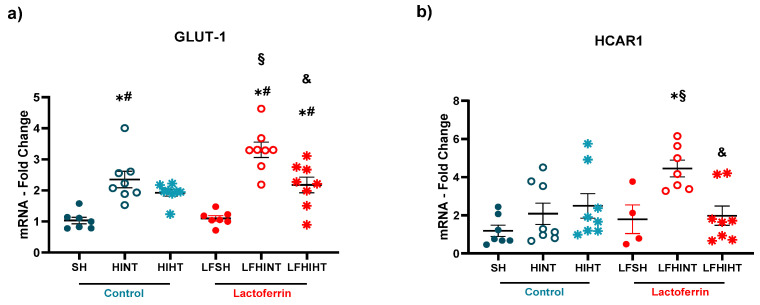
Transcription levels of the genes involved in the glucose (GLUT-1) and lactate (HCAR1) transport in (**a**) and (**b**), respectively in the right hippocampus 24 h after HI. The data are presented as the mean ± SE. A one-way ANOVA was followed by Tukey’s *post hoc* * vs. SH, ^#^ vs. LFSH, ^§^ LFHINT vs. HINT, ^&^ HIHT vs. HINT.

**Figure 11 ijms-24-15583-f011:**
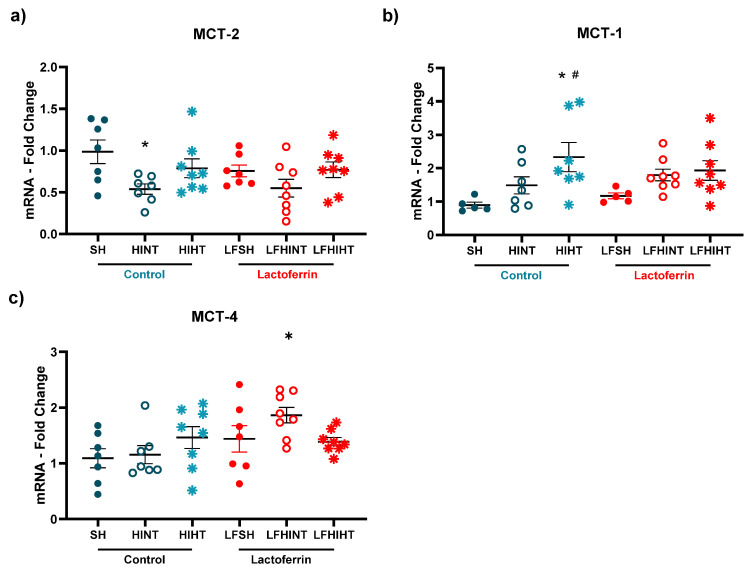
mRNA levels of the genes that the monocarboxylate transporters expressed in the neurons (MCT-2, in (**a**)) and glial cells (MCT-1 and MCT-4 (**b**) and (**c**), respectively)) in the right hippocampus 24 h after HI. The data are presented as the mean ± SE. A one-way ANOVA was followed by Tukey’s *post hoc* * vs. SH, ^#^ vs. LFSH.

**Figure 12 ijms-24-15583-f012:**
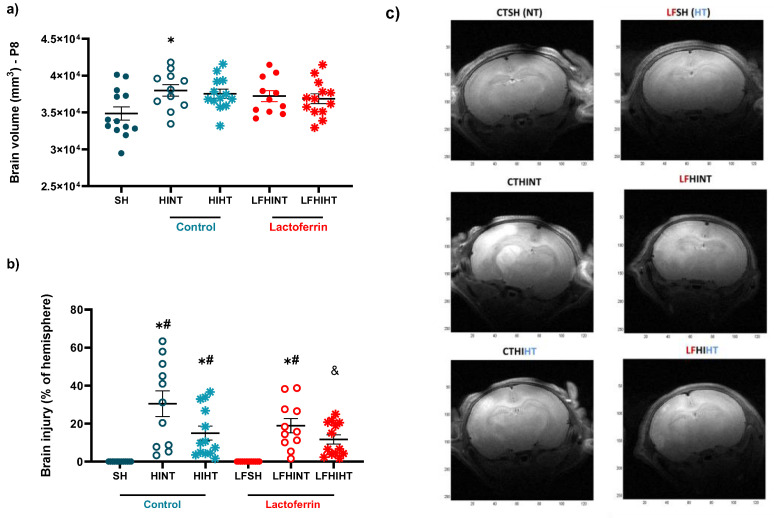
HI injury caused brain swelling and tissue damage. The combination of HT and LF reduced the hyperintense signal to a greater extent 24 h after HI. The brain volumes (**a**) and damages (expressed as percentages of the hemisphere) (**b**) based on the hyperintense T2WI signal 24 h after HI. The representative T_2_W images acquired using a 9.7 T magnet (**c**). The sham animals (HT and NT) were analyzed together since no differences were observed. The data are presented as the mean ± SE. A one-way ANOVA was followed by Tukey’s *post hoc*, * vs. SH, ^#^ vs. LFSH, ^&^ LFHIHT vs. HINT.

**Figure 13 ijms-24-15583-f013:**
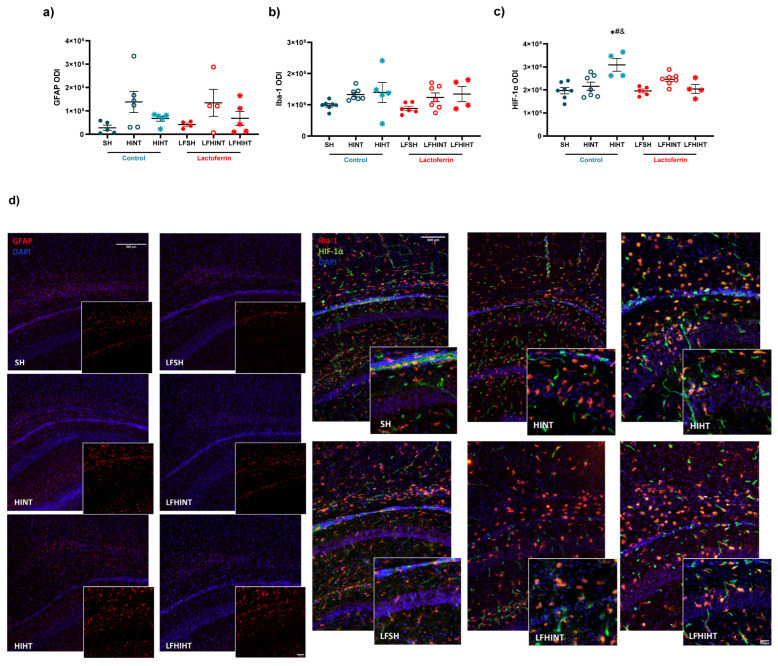
Effects of bovine lactoferrin and hypothermia on the expression of the astrocytes (GFAP, in red—left panels) (**a**), microglia (Iba-1, in red—right panels) (**b**) and the hypoxia-inducible factor (HIF-1α, in green—right panels) (**c**) in the right CA1 24 h after HI. Representative images of the different experimental groups (**d**). In GFAP images scale bar—500 µm; Iba-1 and HIF-1α images scale bar—100 µm. The data are presented as the mean ± SE. A one-way ANOVA was followed by Tukey’s *post hoc* * vs. SH, ^#^ vs. LFSH, ^&^ HIHT vs. HINT and LFHIHT.

**Figure 14 ijms-24-15583-f014:**
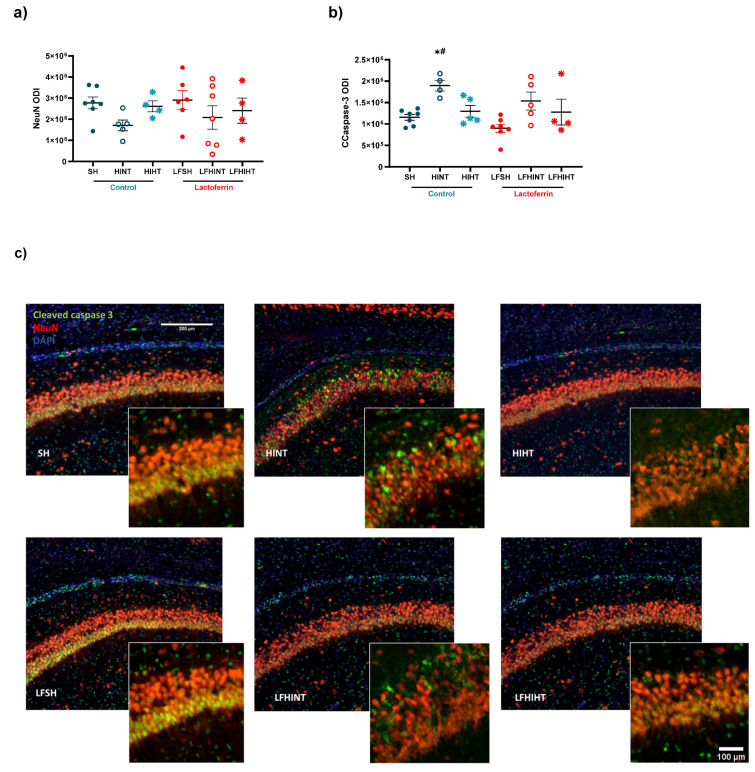
Effects of bovine lactoferrin and hypothermia on the expression of mature neurons (NeuN) (**a**) and cleaved caspase-3 (CCaspase-3) (**b**) in the right CA1 region 24 h after HI. Representative images of the different experimental groups; scale bar 100 µm (**c**). The data are presented as the mean ± SE. A one-way ANOVA was followed by Tukey’s *post hoc* * vs. SH, ^#^ vs. LFSH.

**Figure 15 ijms-24-15583-f015:**
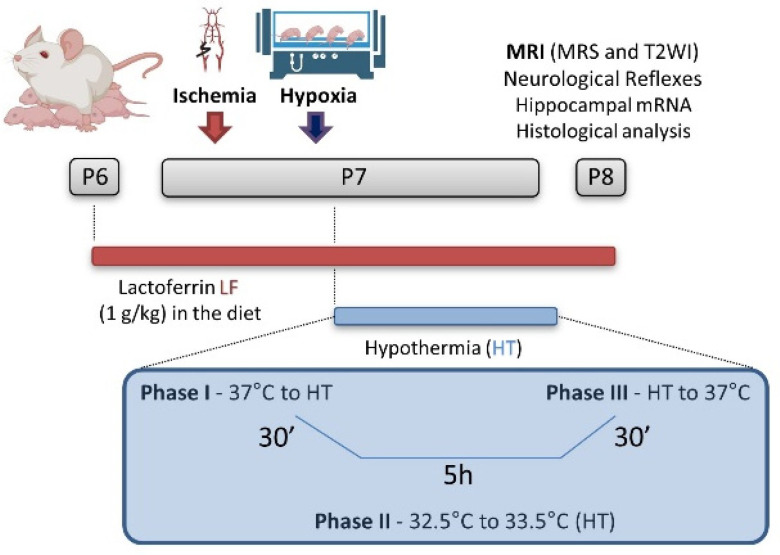
Experimental timeline.

**Figure 16 ijms-24-15583-f016:**
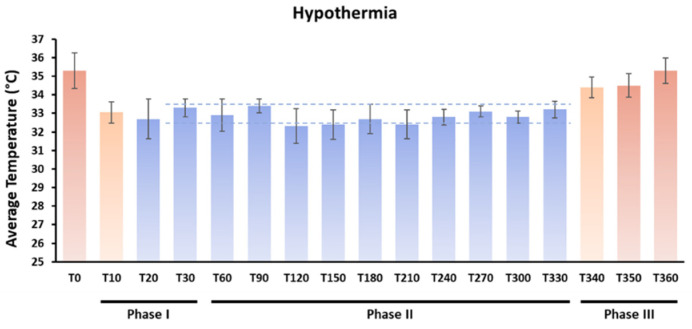
Average temperatures during experimental hypothermia. The dashed lines represent the temperature range (32.5°–33.5 °C) target. The overall average temperature during Phase II (hypothermia) was 32.8 ± 1.88 °C.

## Data Availability

The data that support the findings of this study are available from the corresponding author upon reasonable request.

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
