# Peer review of "Early Neuroprotective Effects of Bovine Lactoferrin Associated with Hypothermia after Neonatal Brain Hypoxia-Ischemia in Rats"

_ijms, 2023, doi:10.3390/ijms242115583_

Round 1

Reviewer 1 Report

Dear authors,

Congratulations for your contributions and for good perspectives for further studies!

For immunofluorescent studies / photos, I would recommend higher magnification (not only x10) as well to mark different colours of fluorescence in every cell type presented on slides.

Best regards!

Author Response

Authors thank the acknowledgement.

The images were improved and we believe the modifications made, make it suitable for the journal standards.

Improvements were made in order to address the comments raised also by reviewers 2 and 3.

Reviewer 2 Report

Line 40: considered "an" etiological; rather than considered etiological. Line 56: close gap between "to" and "its". Line 58: "rodents" rather than rodent. Line 69-70: effectiveness "of" decreasing.  Line 71:  and greatly benefit "by" adjuvant therapies. Line 74: properties have "been" shown.  Line 75: "by" reducing:  Line 79: member of "transferrins" family. Line 80: protecting "the " developing brain. Line 89: name of this study "is" to test..... . Line103: regarding "a" righting "reflex" Line361: delete "been".

Author Response

Authors thank the comments on the manuscript.

Modifications were made accordingly reviewer askings.

Reviewer 3 Report

The present paper provides data to support the hypothesis that LF (lactoferrin) and HT (hypothermia) may act in synergy to increase neuroprotection in a model of neonatal hypoxic-ischemic encephalopathy (NHI). The question addressed is interesting and the experimental protocol is clear and standardized. The present paper follows a dose response paper by the same authors showing that LF treatment reduces neurodegenerative processes in NHI animal model followed up to 25 days. I wonder why the combined effects was not tested for the same time and may be longer, I think it is important to clarify this point. Why it was chosen day 6 to start supplement with LF? Presentation of results might be improved. Images are difficult to read.

Discussion is too long and not well focused additive, disjunctive or competitive effects are mixed. The last sentence try to clarify several points of interest

Author Response

Following the encouraging results presented here, we have conducted experiments evaluating animals at P25, however due to the amount of data, we opted for dividing the results into 2 manuscripts (one focused on early mechanisms), and the second focusing on behavioral and histological outcomes at a later timepoint. Further data should be presented soon.

LF protocol start (P6 - in the evening) was just a matter of having LF available for the pups at the moment of HI (P7), but not for so long that could have altered the neurodevelopment trajectory or having a “preventive” effect, since HIE events are (in its majority) unpredictable events.

Graphs were changed and we hope the modifications have made it clearer for the readers.

We have altered the discussion, but we believe that shortening it could have made harder for the readers to follow the rationale (and our interpretation of the results) since we describe several distinct mechanisms of injury that can be altered by both therapies simultaneously or used alone. 

Round 2

Reviewer 3 Report

revisions and comments are ok